# Evolution of a ZW sex chromosome system in willows

Nan Hu [1], Brian J. Sanderson [2], Minghao Guo [1], Guanqiao Feng[1], Diksha Gambhir[1], Haley Hale[3], Deyan Wang[4], Brennan Hyden[5], Jianquan Liu[4], Lawrence B. Smart [5], Stephen P. DiFazio [6], Tao Ma[4] & Matthew S. Olson [1] ✉

Transitions in the heterogamety of sex chromosomes (e.g., XY to ZW or vice versa) fundamentally alter the genetic basis of sex determination, however the details of these changes have been studied in only a few cases. In an XY to ZW transition, the X is likely to give rise to the W because they both carry feminizing genes and the X is expected to harbour less genetic load than the Y. Here, using a new reference genome for *Salix exigua*, we trace the X, Y, Z, and W sex determination regions during the homologous transition from an XY system to a ZW system in willow (Salix). We show that both the W and the Z arose from the Y chromosome. We find that the new Z chromosome shares multiple homologous putative masculinizing factors with the ancestral Y, whereas the new W lost these masculinizing factors and gained feminizing factors. The origination of both the W and Z from the Y was permitted by an unexpectedly low genetic load on the Y and this indicates that the origins of sex chromosomes during homologous transitions may be more flexible than previously considered.

Sex chromosomes represent one of the most functionally and evolutionarily consequential regions of the genome[1], so their movement is enigmatic. Transitions in sex chromosomes are more common than once thought, and they span the range of possibilities[2–4]. Most studies of the genomic changes during these transitions have focused on homogametic transitions, which are defined as those that maintain the same heterogametic sex (XY to XY, or ZW to ZW)[1,5–14]. Heterogametic transitions (XY to ZW, or ZW to XY) are less common than homogametic transitions[15–19], and are often more genetically complex, requiring a shift in the relative dominance of the masculinizing and feminizing chromosomes. Tracing the fate of the old chromosome pairs into the new pair allows extraordinary insight into the changes in genetic components and is a key step toward understanding how and why these transitions occur.

Sex chromosome transitions also can be categorized by whether the sex determination loci are on different linkage groups or result from a turnover associated with the same linkage group[20]. During a heterologous transition, wherein the sex determination regions are on different chromosomes (or linkage groups), the old sex chromosome (or linkage group) converts back to an autosome. However, during a homologous transition, wherein a new sex determination allele (e.g., W) arises in the same location as the ancestral sex determination alleles (e.g., XY), an unstable three allele polymorphism (e.g., X, Y, and W chromosomes) evolves into a two allele polymorphism (e.g., ZW) During these homologous transitions, a naive hypothesis postulates that the Z is derived from Y (or vice-versa) because both chromosomes are associated with masculinization and that the W is derived from the X because both are associated with feminization[21]. This pattern is supported in the hybrid populations of Japanese soil frogs where the neo-W chromosome originated from the X and the Z arose from the Y[22,23]. However, in other taxa such as coenophilid snakes, which exhibit ZW to XY transitions, shared ancestry among several genes on the X, Y,

[1]Department of Biological Sciences, Texas Tech University, Lubbock, TX, USA. [2]Department of Molecular Biosciences, University of Kansas, Lawrence, KS, USA. [3]HudsonAlpha Institute for Biotechnology, 601 Genome Way, Huntsville, AL, USA. [4]Key Laboratory of Bio-Resource and Eco-Environment of Ministry of Education, College of Life Sciences, Sichuan University, Chengdu, China. [5]Horticulture Section, School of Integrative Plant Science, Cornell University, Cornell AgriTech, Geneva, NY, USA. [6]Department of Biology, West Virginia University, Morgantown, WV, USA. ✉e-mail: matt.olson@ttu.edu

and Z chromosomes suggested that both the X and Y were evolved from the Z, and the W was lost during the transition[24]. Finally, recent studies in *Populus* section *Populus* indicate that a complex series of events during the shift from ZW to XY resulted in the W converting into the Y, but the origin of the X remains unclear[25,26].

The two largest genera in the *Salicaceae*, *Populus* and *Salix* within which all but one species (*S. martiana*) are dioecious[27,28], have recently been discovered to have been influenced by multiple sex chromosome transitions in both directions between XY and ZW[29,30]. In *Populus*, XY sex determination systems have been reported on both chromosomes 14 and 19[26,31–34] and different ZW systems have been identified on chromosome 19 in two species within the same subgenus[25,26]. In *Salix*, XY systems have been identified on chromosomes 7 and 15, and a ZW system was identified on chromosome 15, but the ancestral states and direction of change remain in dispute[29,35–38]. In *Populus*, the genetic basis of sex determination was recently discovered to be affected by complex interactions between a cytokinin response regulator (*RR17*) and small interfering RNAs (siRNAs) that are generated by partial repeats of *RR17* (hereafter 'partial duplicates')[26,33]. Expression of the full length *RR17* gene results in a female phenotype[26]. In ZW species of *Populus* the intact *RR17* genes are present only on the W chromosome, and their absence in ZZ results in male phenotypes[26]. In XY species, complete *RR17* gene copies are present in the pseudo-autosomal regions on both the X and the Y chromosomes[26]. Partial duplicates are present only on the Y and generate siRNAs that block the expression of the intact copies of *RR17* in males via RNA-directed DNA methylation, thereby acting as a dominant masculinizing factor[26]. The details of sex determination appear to vary between *Populus* and *Salix*[39], although the fundamental associations of *RR17* with feminization is present in *Salix*, suggesting common origins of the sex determination mechanism throughout both genera[26,30].

Here we trace the X, Y, Z and W sex determination regions during the homologous transition from an XY into a new ZW system sex determination region on chromosome 15 in *Salix*. We start by documenting the chromosomal location and heterogamety of the sex determination system in *S. exigua* using a new reference genome built for this species. We show that *S. exigua* is in a key grade on the phylogeny, which confirms the direction of the 15XY to 15ZW shift that occurred during diversification of the genus. We then characterize one large contig as unique to the Y chromosome, which carried several small partial repeats of the *RR17* gene that are unique to the *S. exigua* male genome. Finally, using a half-sib family, we identify a suite of polymorphic homologs that are present on the X and Y of *S. exigua* and the Z and W of *S. purpurea*, resulting in the ability to trace the loss of the X sex determination haplotype during this transition. These results provide insight into chromosomal changes during a homologous shift in the heterogametic status of sex chromosomes allowing illumination of the factors influencing these transitions.

## Results

### Genome assembly and a 15XY SDR in *Salix exigua*

Using Oxford Nanopore reads combined with Hi-C scaffolding, we developed a chromosome-level assembly for a *Salix exigua* male (XY; N50 of 10.3 M, 122.5x coverage) to compare with the previously-assembled *S. purpurea* (ZW) genome[29]. 89.5% of the nanopore contigs were anchored onto 19 chromosomes. The reliability of the *S. exigua* assembly was supported by the strong synteny with the *S. purpurea* genome (Fig. S1), the independence of chromosomes on the Hi-C interaction map (Fig. S2), and a BUSCO score of 98.1%. A female *S. exigua* genome (XX) also was assembled using only Nanopore and Illumina sequencing and consisted of 1650 contigs with an N50 of 665k (91.4x coverage). Eighteen contigs comprise a preliminary X chromosome assembly based on synteny with the male Chr15 assembly.

Using a custom sequence capture array targeting exon-anchored sequences[29], we genotyped 2,023,175 SNPs in 24 males and 24 females

of *S. exigua*, which was narrowed down to 267,938 SNPS after hard filtering. Using the male *S. exigua* reference genome, a GWAS on male and female phenotypes using a conservative Bonferroni-adjusted critical value revealed 20 SNPs that were strongly and significantly associated with sex (Fig. 1a). These 20 SNPs identified a homologous region between the X- and Y- sex determination regions (SDRs) that was located between 1.8 and 1.9 Mb on Chr15 (Fig. 1a, c; Supplementary Data 1). For these 20 loci, average heterozygosity in males was 91.6% and average homozygosity in females was 96.6%, indicating an XY male heterogametic sex determination system (Fig. 1b). To identify the location of this region in *S. purpurea*, we conducted another GWAS after re-calling SNPs against the *S. purpurea* reference genome with the 15 W (and without the 15Z) chromosome. We identified 194,250 SNPs from the same 48 individuals, and 66 of these SNPs were significantly associated with sex (Fig. 1d). These 66 SNPs mapped to a 3.3 MB region (4.3 MB–7.6 MB) that overlapped with the SDR[35] on Chr15W of *S. purpurea* (Fig. 1d).

### A Y-specific contig on the male Chromosome 15

We further identified a hemizygous portion of the *S. exigua* SDR on that displayed greatly reduced read depth in females compared to males (Fig. 2a). Contig 511 from the male Nanopore raw assembly uniquely assembled to this same portion of the sex determination region between 0.3 MB and 1.2 MB on the unphased Chr15 male assembly (Fig. 2b; Supplementary Data 2). Many fewer SNPs were called for females than for males in this region (Fig. 2c), and based on our lastz alignment, we found no synteny between contig 511, which aligned between 0.6MB to 1.2MB, to any portion of the female *S. exigua* (XX) genome assembly (Fig. 2b). Since the preponderance of evidence indicated that contig 511 is male specific and in the SDR, we assigned it as a Y-specific region of *S. exigua*. Contig 511 exhibited partial synteny with a large section of the sex determination region of *S. purpurea* Chr15Z and less synteny with *S. purpurea* Chr15W (Fig. 2b, Supplementary Data 3). Overall, we estimate that the SDR in *S. exigua* is approximately 1.3 MB and extends from 0.6 to 1.9 MB of Chr15 on our unphased hybrid X-Y assembly (Fig. 2b).

### The Z and W chromosomes originated from the Y

The nested structure of the species phylogeny that included two species with all three sex chromosome types in *Salix* (15XY, 7XY[29], and 15ZW[36, 40]; Fig. 3) revealed that the 15ZW chromosome was derived from an ancestor with the 15XY chromosome (a 15XY to 15ZW transition, Fig. 3). To further understand the chromosomal changes during the shift from XY to ZW, we identified 30 alleles on the Y chromosome in *S. exigua* by comparing SNPs from a female to 48 of her half-sib progeny (Fig. 1d, Supplementary Data 4). The Y-specific alleles in 28 of these 30 exonic regions in *S. exigua* were identical to alleles in the Z-SDR of 10 *S. purpurea* males (ZZ) and the W-SDR alleles from the *S. purpurea* reference genome (Fig. 4a). In contrast, 0 of the 30 X-specific alleles (anchored to exons) from *S. exigua* were present on either the 15Z-SDR or 15W-SDR of *S. purpurea*. The different relative locations of these exonic regions along the SDR in *S. exigua* and along the Z- and W-SDRs of *S. purpurea* indicate a complex history of transversions, inversions, insertions, and deletions associated with the sex chromosome turnover event (Fig. 4a, S3). 26 of these 30 SDR-associated SNPs were associated with exons (Table 1). A phylogeny of concatenated exons that contain these 30 SNPs showed that Z, W, and Y haplotypes are monophyletic with respect to the X-regions (Fig. 4b) indicating that the Z and W haplotypes in the SDR of *S. purpurea* were derived from an ancestral-Y chromosome and the X-haplotypes were lost during the transition from XY to ZW.

### RR17 genes and partial duplicates

Two homologs to the type A cytokinin response regulator 17 (*RR17*) were identified in both the male and female *S. exigua* genomes and

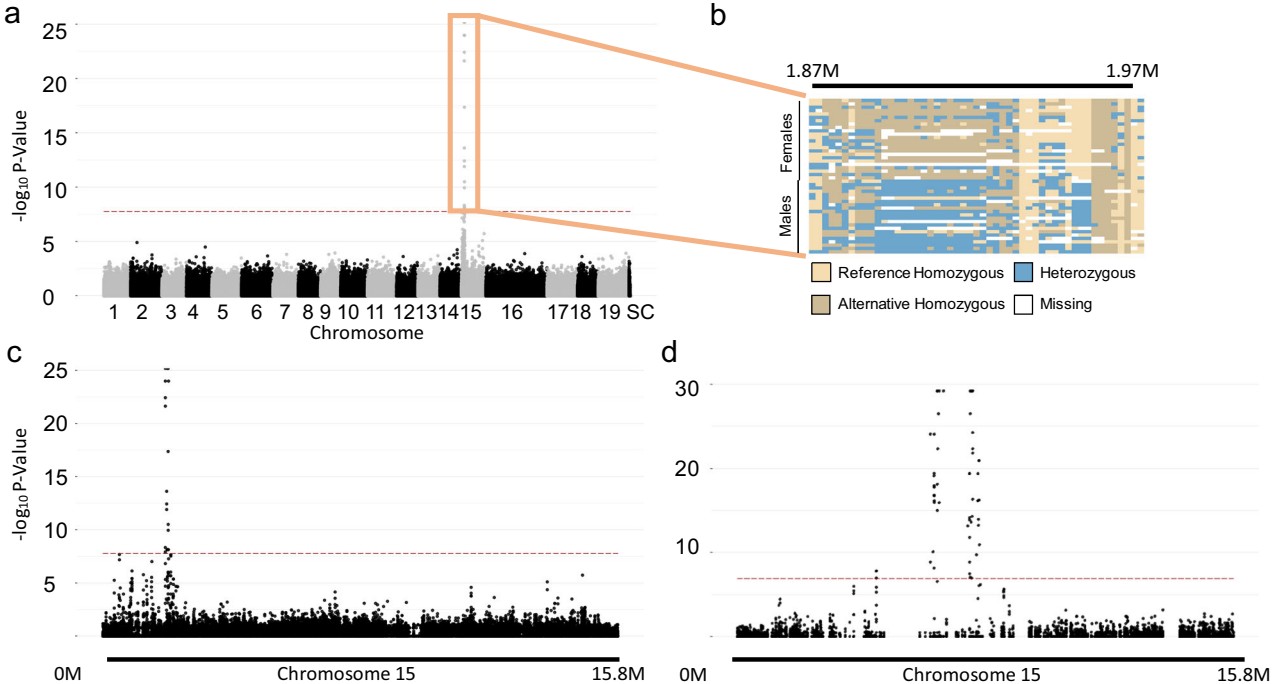

**Fig. 1 | Mapping sex-associated SNPs in *S. exigua*. a** GWAS results show sex is strongly associated with Chr15 in *S. exigua* when *S. exigua* is used as the reference genome. The X-axis indicates the chromosomal position and SC refers to unassembled scaffolds. The Y-axis represents the significance of the association. The red dashed line is the significance threshold after using Bonferroni multiple testing correction (one sided α < 0.05). Each dot in the figure represents a single SNP after filtering. **b** Male genotypes of SNPs around the SDR were primarily heterozygous and females were homozygous indicating an XY sex determination system. **c** A magnification of the GWAS results for Chr15 in *Salix exigua*. **d** GWAS results for *S. exigua* genotypes from the same individuals as in 1a when the *S. purpurea* Chr15W chromosome is used as the reference genome. Red dashed lines represent the same significance cutoffs as in part (**a**).

were mapped to Chr19 in our complete male Hi-C assisted assembly (Supplementary Data 5). In addition, we identified 5 small partial duplicates of the *RR17* gene on contig 511 (Supplementary Data 6). Partial duplicates were absent from the remainder of the male genome and the entire the female genome, including Chr15W. Several patterns indicate homology between these 5 partial duplicates and some of the 14 previously-identified partial duplicates on the *S. purpurea* Chr15 Z[30]. First, three of the homologs, Se2, Se4, and Se5, were the exact same lengths and had > 86% similarity with two partial duplicates found on *S. purpurea* Chr15Z (Fig. 5a). The two other partial duplicates differed between *S. exigua* and *S. purpurea* had > 77% similarity in the regions of overlap but differed in their start and end positions (Fig. 5a; Supplementary Data 6). Second, the direction and arrangement of these five shared partial duplicates was the same in both the Y (*S. exigua*) and Z (*S. purpurea*) chromosomes (Fig. 5b). Finally, a phylogenetic comparison of homologous regions from each pair of partial duplicates and the homologous regions on the intact *RR17* genes on Chr15W and Chr19 supported a single origin for each partial duplicate that occurred prior to the divergence between *S. exigua* and *S. purpurea* (Fig. 6). Interestingly, two of the copies with identical lengths (Se4-Sp6 & Se5-Sp7) exhibited high divergence from the ancestral full-length copy, whereas the other 3 pairs of partial duplicates (Se1-Sp2, Se2-Sp4, & Se3-Sp5) exhibited less divergence from the ancestral full-length copy (Figs. 6, S4), indicating rapid evolution of Se4-Sp6 and Se5-Sp7 after their origination.

## Discussion

Based on a combination of phylogeny and sex determination region (SDR) mapping, we identified a homologous transition from a male heterogametic (XY) to a female heterogametic (ZW) sex determination system on chromosome 15 in willows. During this transition, both the W and Z chromosomes were derived from an ancestral Y chromosome, and the X was lost. Many components in the sex determination region

were retained during this transition as detailed in Fig. 7a. Prior to the transition, at least five duplicates of small segments of the *RR17* gene ('partial duplicates') were present on the Y chromosome. The function of these *RR17* partial duplicates in *Salix* is still under investigation[26,35,37]; however, inverted repeats of *RR17* partial duplicates regulate sex determination via siRNA intermediates in *Populus tremula*[25,26], which is in the sister genus to *Salix*, and *RR17* partial duplicates in *Salix arbutifolia*, which also has a 15XY sex determination system, generate siRNAs[25,30,41]. Interestingly, there is currently no evidence of siRNA expression of *RR17* partial duplicates in *S. purpurea*, so this masculinizing component of the XY system may have been lost or modified in the transition to ZW[39]. After the divergence of ancestors of *S. exigua* and *S. purpurea*, a new W chromosome evolved in the *S. purpurea* lineage coincident with four complete copies of *RR17* being duplicated and translocated from Chr19 to the sex determination region of Chr15[35]. A previous collinearity analysis between Z and W in *S. purpurea* showed that the four full-length copies of *RR17* are located within two palindromes on the W chromosome in a region that differs from the Z by a large inversion (Fig S3)[35] and indicates a large-scale chromosomal rearrangement during the evolution of the W chromosome. *RR17* was found to have extremely female biased expression in *S. purpurea*, which may serve as a feminizing gene similar to its function in *Populus*[35]. Based on the function of *RR17*, it has been hypothesized that the new sex determination system incorporates a dosage mechanism associated with multiple *RR17* repeats[35,39,42]. After the W chromosome arose, a polymorphism with W, X, and Y chromosomes was segregating in an ancestral population (Fig. 6a). This polymorphism was not stable, as the X chromosome was ultimately lost, and a new ZW female heterogametic system spread to fixation. An experimental crossing study between *S. exigua* and *S. eriocephala*, which is closely related to *S. purpurea* and likely harbors a 15ZW system, supports some aspects of our model. In this work[43], F1 progeny from a cross between a female *S. exigua* (XX) and a male *S. eriocephala* (putative ZZ) were all males

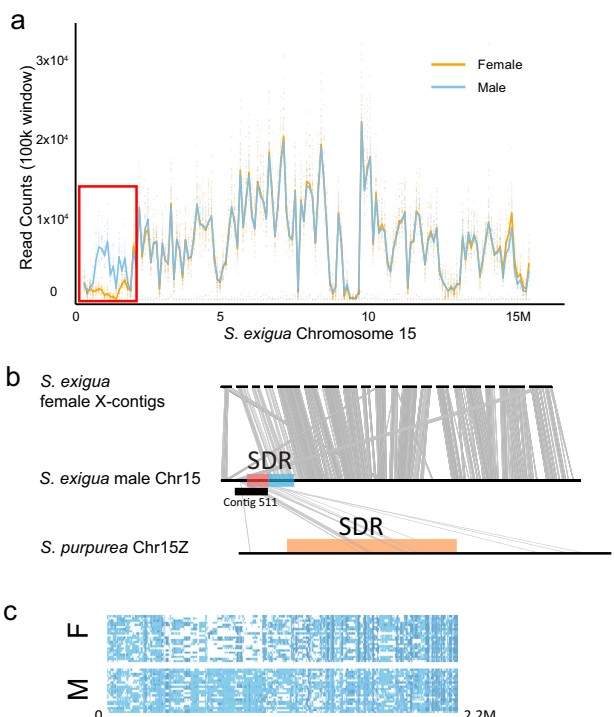

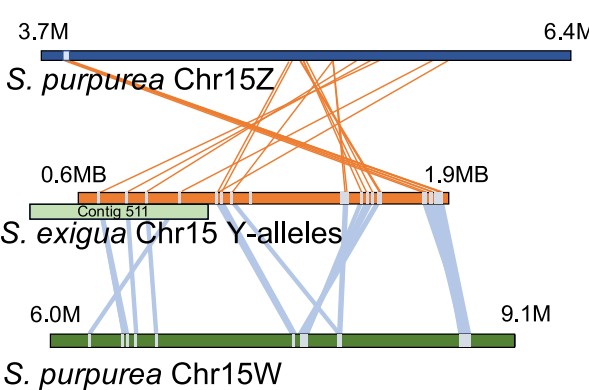

**Fig. 2 | The X contigs and the Y-specific contig in *S. exigua*. a** Mapping read (MQ > 30) depths between 24 males and 24 females on *S. exigua* Chr15. The female genotypes had low depth between 0.6 and 1.3 Mb on Chr15, are highlighted by the red box, and indicate a Y-specific assembly on Chr15. The blue line and dots refer to male average and individual read depth, respectively. The orange line and dots refer to female average and individual read depth, respectively. **b** Alignments among the *S. exigua* X and Y chromosomes and the *S. pururea* Z chromosome. Grey lines represent syntenic genes among the *S. exigua* X contigs, the *S. exigua* male Chr15 (XY), and the *S. purpurea* Chr15Z. The location where contig 511 assembles is shown as a black bar. The red portion of the sex determination region (SDR) in the S. exigua male is the Y-specific region and is associated with a portion of contig 511, the blue portion of the SDR exhibits homology between the X and Y chromosomes. The SDR on *S. purpurea* Chr15Z is represented by an orange bar. **c** SNP presence between 0 and 2.2 Mb showing a low frequency of SNP calls in females between 0.6 and 1.3 Mb. Blue dots refer to the presence of SNP.

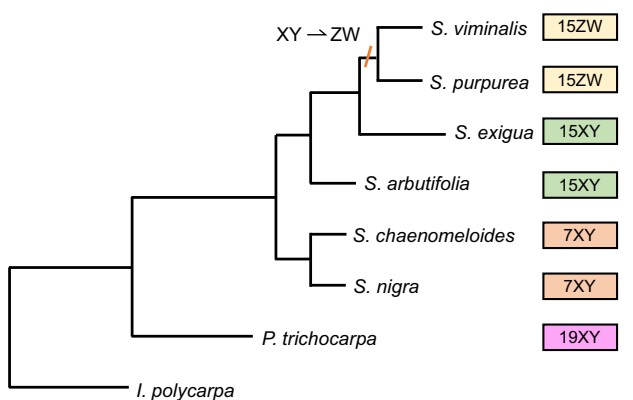

**Fig. 3 | Sex chromosome transitions in *Salix*.** The phylogeny of 8 species in the *Salicaceae* family shows how the sex chromosomes have changed across the phylogeny and where the sex determination system shifted from XY to ZW (Adapted from Sanderson et al. 2023[75]).

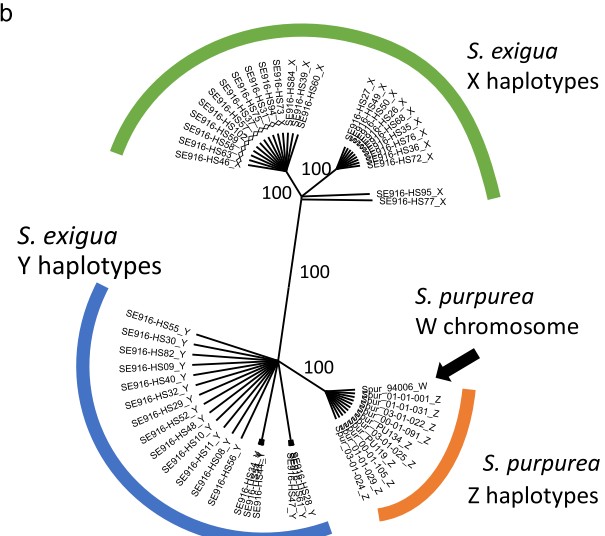

**Fig. 4 | Relationships among 30 exon-anchored homologous regions on the Z, Y, and W sex determination regions. a** Alignment of the exon regions (grey bars) with Y-associated SNPs across *S. exigua* and *S. purpurea*. The relative location of contig 511 is shown in green. **b** A phylogeny based on the concatenated exon regions shows that the Z and W chromosomes are derived from an ancestral Y. Bootstrap support > 70 is indicated.

(putative XZ), indicating the Z is functionally similar to the Y chromosome, with both being masculinizing and dominant to the X. Moreover, the reciprocal cross (putative ZW x XY) generated both males (putative XZ and YZ) and females (putative WX and WY),

suggesting the W is feminizing and dominant to both the X and Y chromosomes.

The evolution of a W chromosome from an ancestral-Y has not been documented in other heterogametic sex chromosomes transitions. Theoretically, there are fewer barriers to the transition when the W is derived from the X than from the Y. For instance, both the X and W are associated with feminization, whereas the Y chromosome is associated with masculinization[44], so the function of the W must shift if it is derived from the Y. Thus, for a Y to develop into a W, it requires three changes: 1) the masking or loss of masculinizing function, 2) the gain of feminizing function, and 3) the development of dominance over the Y chromosome. Notably, this is made simpler when the first two changes result from the same mutation[44]. Alternatively, for an X to develop into a W, only one change, from recessive into a dominant action over the Y chromosome, is required. The origination of the W from the X was concluded in Japanese soil frogs, where the X and W share a female-determining gene and nearly identical structure[23]. Another impediment to the W being derived from the Y is that the Y chromosome usually has a lower effective population size and higher genetic load compared to the X[21]. During the transitional stage, the polymorphism with X, Y ( = Z), and W sex determination regions (SDRs) results in the

**Table 1 | Genes with phased SNPs that were used to follow the transitions between X, Y, Z, and W haplotypes**

| SNP position on *S. purpurea* Chr15 | Annotation Gene in *S. purpurea* phytozome v5.1 | Annotation in *A. thaliana* | Functional Annotation |
|---|---|---|---|
| 6278738, 6278768 | 15ZG056600 | AT5G23090.1 | nuclear factor Y, subunit B13 |
| 6277500 | 15WG057600 | AT3G18550.2 | TCP family transcription factor |
| 6341283 | 15WG058100 | AT1G49050.1 | Eukaryotic aspartyl protease family protein |
| 6468236 | 15WG058800 | AT1G25440.1 | B-box type zinc finger protein with CCT domain |
| 7308166, 7308168 | 15WG064300 | AT5G61280.1 | Remorin family protein |
| 7334961 | 15WG064500 | AT5G61210.1 | soluble N-ethylmaleimide-sensitive factor adaptor protein 33 |
| 7338400, 7338458 | 15WG064600 | AT5G07890.3 | myosin heavy chain-related |
| 7352217 | 15WG064700 | AT5G07910.1 | Leucine-rich repeat (LRR) family protein |
| 7373897 | 15WG064900 | AT5G60880.1 | breaking of asymmetry in the stomatal lineage |
| 7387078, 7387090 | 15WG065000 | AT2G03810.4 | 18 S pre-ribosomal assembly protein gar2-related |
| 7580404, 7580406, 7582930 | 15WG065600 | AT1G18485.1 | Pentatricopeptide repeat (PPR) superfamily protein |
| 6059052 | 15WG056500 | AT1G18470.1 | Transmembrane Fragile-X-F-associated protein |
| 7289474 | 15WG064200 | AT1G61415.1 | Unknown |
| 8352593 | 15WG070000 | AT1G18520.1 | tetraspanin11 |
| 7289429 | 15WG064200 | AT1G61415.1 | Unknown |
| 8312500, 8312503, 8312508 | 15WG069700 | AT5G07980.1 | dentin sialophosphoprotein-related |
| 8326228 | 15WG069800 | AT5G23050.1 | acyl-activating enzyme 17 |
| 8345826 | 15WG069900 | AT1G18520.1 | tetraspanin11 |
| 8359542 | Non-genic region | | |
| 8359191 | Non-genic region | | |
| 8359477 | Non-genic region | | |
| 8346031 | Non-genic region | | |

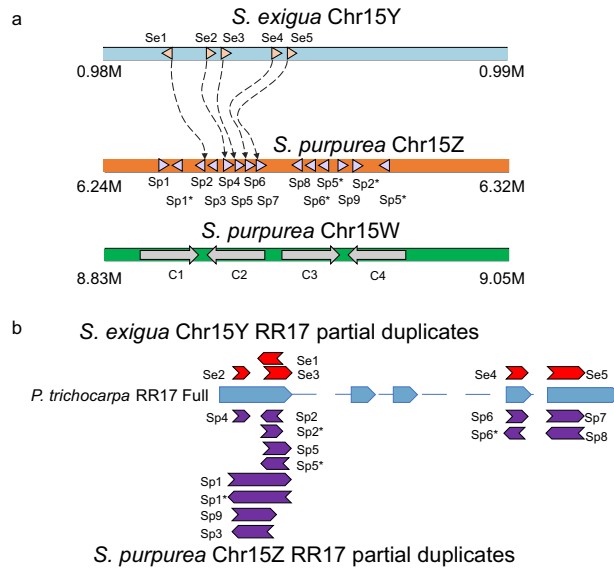

**Fig. 5 | *RR17* genes and partial duplicates in *S. exigua* and *S. purpurea* chromosome 15. a** Positions (triangles) and copy names of *RR17* genes and partial duplicates on the *S. exigua* Y, *S. purpurea* Z, and *S. purpurea* W chromosomes. Triangles are pointed in the direction of the sequence relative to the original *RR17* gene (XM_024590879.1). Dashed lines show the homology between *RR17* partial duplicates on the Y and Z chromosomes. Large arrows on the W chromosome indicate full length intact *RR17* copies. **b** Homology of *RR17* partial duplicates relative to the intact *RR17* gene in *Populus trichocarpa*. Red arrows represent partial duplicated of *RR17* in *S. exigua* and are positioned according to their homology with the full length intact *RR17* gene (Blue boxes). Purple arrows represent copies in *S. purpurea* positioned with their homology relative to the intact copy of *RR17*. * indicates the duplicated palindromes within 10 bp in length.

production of two deleterious homozygous genotypes when the W is derived from the Y (YY and YW; Fig. 7b) and only one deleterious homozygous genotype (YY) when the W is derived from the X (Fig. 7b). Interestingly, both overcoming multiple mutational effects and the potentially higher genetic load in the Y were seemingly evaded during the XY to ZW transition in *Salix*. In support, in *S. arbutifolia*, which also is 15XY, complete sequencing of a population sample of SDRs on the Y chromosome revealed that they did not harbor more genetic load than either the SDR on the X or the pseudo-autosomal region (PAR) of chromosome 15[30].

We confirmed a homologous transition from XY to ZW transition in *Salix*. It appears that the ZW sex determination system in all of the economically important subgenus Vetrix (shrub willows) has been derived from this transition because all of the studied species in this subgenus have a 15ZW sex chromosome system[36,38,40]. Combined with the previous discovery of an XY to ZW transition between lineages of *P. tremula* and *P. alba*, shifts in heterogamety occurred at least twice in *Salicaceae*[26]. In other dioecious angiosperm clades, shifts in heterogamety are rare[45]. One exception can be found in *Silene* section *Otites*. Here, sex chromosome systems transitioned from ZW to XY and then back to ZW, which was perhaps permitted by the lack of strong degeneration among the sex chromosomes[46]. A similar lack of strong degeneration in *Populus* and *Salix* sex chromosomes may also contribute to the frequency of shifts in *Salicaceae*.

The mechanism driving the XY to ZW shifts in *Salix* remains unresolved, but no current model strongly aligns with our observations in *Salix*. Models based on linkage of the sex determination genes with sexual antagonistic loci are possible, but more difficult to imagine in plants than in animals. In cichlid fish, for instance, color polymorphisms are sexually antagonistic, because females do not benefit from the same color patterns that males do[13]. However, in animal pollinated dioecious plants like willows, one would hypothesize that male and female flowers must both attract the same pollinators to

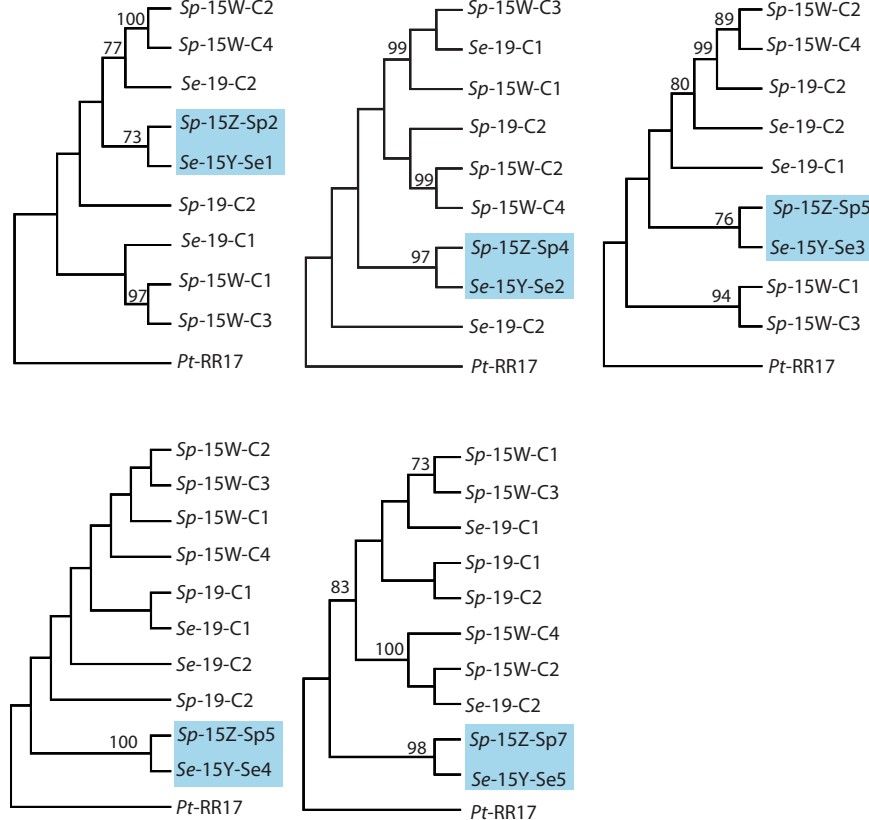

**Fig. 6 | Homologous partial duplicates on the Y and Z chromosomes each have a single origin.** Phylogenies include homologous partial duplicates (highlighted in blue) on *S. exigua*, *S. purpurea* Z, and homologous regions on the intact copy of *RR17* in *S. exigua*, *S. purpurea*, *and P. trichocarpa*. The patterns suggest that the partial duplicates on Z and Y each have a single origin. Bootstrap support > 70 is indicated. The tip labels are formatted as Species-Chromosome-Copy number. Sp: *S. purpurea*; Se: *S. exigua*; Pt: *P. trichocarpa*. C refers to the intact copies of *RR17*.

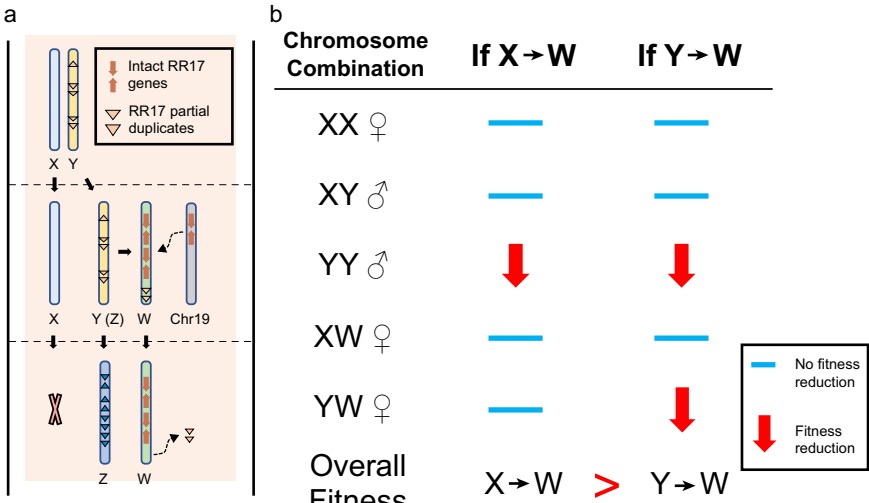

**Fig. 7 | The fate of sex chromosomes during the transition from XY to ZW sex systems. a** Model of sex determination evolution from XY to ZW in *Salix*. The X was lost during the transition while the Y developed into the Z and W. Translocations from the Chr19 copy introduced four copies of full length *RR17* genes into the new W chromosome, and *RR17* partial duplicates were deleted from the W. The Y and Z further differentiated by the loss (not shown) or gain (shown) of additional partial duplications of *RR17*. **b** Comparisons of the fitness of genotypes during the X, Y, W polymorphic transition assuming the Y carries genetic load under two scenarios: if the W was derived from the X or from the Y. If the W was derived from the X, and only Y carries a mutation load, YY individuals would have low fitness due to homozygous deleterious mutations. If W was derived from the Y, both the YY and the YW individuals would suffer from the homozygous genetic load.

assure pollination, limiting the extent of sexually antagonistic dimorphisms associated with pollinator attraction and mating[47]. Nonetheless, the emissions of volatile herbivore defense compounds is strongly dimorphic between male and female flowers of *S. purpurea*[48]. Also, both *S. nigra* and *S. exigua* males produce more flowers per catkin and more catkins than females and may form bases for sexual antagonism. Mechanisms related to the hot-potato model[49] also do not appear to have strongly contributed to the heterogametic transition in *Salix*. In this model, heterogametic transitions are unlikely because strong selection against the mutation-loaded Y hinders the spread of an epistatically dominant female sex-determining locus. For example, in true frogs, where build-up of genetic load on the Y is thought to influence sex chromosome transitions, 11 out of 13 transitions preserved the ancestral male heterogamety[14]. Meiotic drive can also induce sex chromosome transitions when a new sex determining gene restores a 1:1 progeny sex ratio[50]. In this process, genetic load is expected to develop when strong meiotic drive persists, and genetic load was not observed on the 15Y chromosome of a close relative of *S. exigua*[30]. Also, homologous heterogametic transitions may be less likely induced by meiotic drive because the loci to restore the sex ratio need to arise on chromosomal regions unlinked to the sex determination region[50]. Notably, however, female sex ratio bias has been reported in several species in subgenus Vetrix[51–55]. Finally, the heterogametic transition in *Salix* may have resulted from solely from genetic drift, although under this model XY to ZW shifts are less common than XY to XY shifts unless extreme polygyny is present[56].

This study exemplifies how studies of sex chromosomes in plants can illuminate a broader understanding of the changes in genomic architecture induced by sex chromosome turnovers[57]. As more transitions are studied in additional systems, it will be instructive to understand whether W chromosomes originating from the Y is a rare event. Homologous transitions inevitably require crossing a different set of evolutionary barriers than heterologous transitions, yet current models describing homologous and heterologous transitions do not provide sufficient detail regarding the evolutionary mechanisms driving these processes[20]. Interestingly, genetic load is often considered to be a key factor in the evolution of sex chromosomes. However, during the transition from Y to W chromosomes, genetic load must be very limited or avoided to ensure that the transition is not counteracted by selection. Future work on understanding how sex chromosomes maintain low genetic load or how genetically loaded chromosomes are retained during heterogametic transitions will provide a better understanding of these processes.

## Methods

### Salix exigua genome assembly
Fresh frozen leaf tissues were collected for genome sequencing and assembled from one male (SE967M) and one female (SE916F) that originated from the Rio Bonito population in eastern New Mexico (collecting permission obtained from main office). High molecular weight DNA was extracted and sequenced by Nextomics Biosciences (Wuhan, China) using a MinIon platform (Oxford Nanopore Technologies, Oxford, UK). For sequencing error correction, 150 bp pair-ended Illumina short-read sequencing was also performed by Nextomics Biosciences (Wuhan, China) on these same individuals using MiSeq (Illumina, Inc., San Diego, CA, US). Finally, for scaffolding Hi-C libraries were constructed for the male individual and sequenced on the MiSeq platform.

The Oxford nanopore reads were filtered for average read quality >= 7 using Guppy (Oxford Nanopore Technologies, Oxford, UK), and adapters were trimmed by Porechop v0.2.4 using default parameters[58]. The raw assembly was completed using Flye v2.9-b1778 with the no scaffolding mode[59], as recommended when scaffolding with Hi-C, and polished with Illumina NGS reads for 4 rounds using Nextpolish v1.3.1[60]. This resulted in an initial assembly N50 of 351 k. Hi-C reads

were aligned with the nanopore raw assembly contigs using Juicer v1.5[61], and contig scaffolding into chromosomes was finished by 3d-dna (version 180922)[62] using diploid mode and correction for 10 rounds to generate the final chromosome-level assembly (N50 = 10.3 M). In total, 122.5 X coverage (49 Gb) of long reads were used to assemble the raw genome of *S. exigua*, and 427.5 X coverage (171 Gb) short reads from Illumina were used for polishing. An all-to-all alignment using lastz v1.1.13 between our *S. exigua* assembly and the *S. purpurea* 94006 PacBio assembly v5.1 (https://phytozome-next.jgi. doe.gov/info/Spurpurea_v5_1) was conducted to confirm assembly quality using the following parameter settings: --chain --gapped --transition --maxwordcount=4 --exact=100 --step=20[63]. Alignments were plotted using last-dotplot[63]. Chromosomes were numbered according to the *S. purpurea* 94006 PacBio assembly v5.1[35]. A female *S. exigua* was assembled from nanopore long reads (49 Gb, 91.4 X) using Flye v2.9-b1778 with self-scaffolding model[59].

### Homologous annotation
Both of our assembly were annotated by applying the annotations from homologous regions of the *S. purpurea* v5.1 genome (JGI). The genomic mapping was done by minimap2[64]. Homologous regions were further annotated using liftoff using default settings[65]. The X chromosome was constructed by identifying contigs from female SE916F that were > 300k in total length and contained a minimal density of genes ( > 0.0001 gene/bp) annotated to either the Chr15Z or Chr15W of *S. purpurea* v5.1 (JGI). Selected contigs were aligned to the male *S. exigua* Chr15 assembly and concatenated separated by 100 N nucleotides as putative gaps.

### Sequence capture and SDR mapping
*Salix exigua* leaf tissue from 24 males and 24 females were collected near the Rio Bonito in Fort Stanton-Snowy River Cave National Conservation Area, Lincoln County, Ruidoso, New Mexico in the spring of 2017 (collecting permission obtained from main office). Tissues were dried in silica gel prior to DNA extraction. Sampled individuals were > 25 m apart to minimize sampling from the same genotype. Sex was scored by the presence of male or female flowers on catkins. DNA was extracted from *S. exigua* leaves following the protocol from the Qiagen Plant DNA Extraction Kit (Qiagen, Hilden, Germany). Bar-coded libraries were made for each individual using the NEB Next Ultra II Library preparation Kit (New England Biolabs, Ipswitch, MA, USA). 60,000 RNA baits were designed from coding regions of *S. purpurea* 94006 PacBio assembly as described in Sanderson et al. 2021 and were used to enrich 16580 target genes in library sample[29] using standard protocols provided by Arbor Biosciences myProbes protocol v 3.0.1. Eight libraries were pooled prior to the sequence capture hybridization. After hybridization, 48 hybridized libraries were pooled and sequenced to 60x on Illumina HiSeq 5000 Genome Analyzer (Illumina, Inc., San Diego, CA, USA). To survey variation in Z chromosomes in *S. purpurea*, leaves were sampled for DNA extractions using a Plant DNA Extraction Kit from a diverse set of 10 males in the germplasm collection of the willow breeding program at Cornell University in Geneva, NY. *S. purpurea* library preparations were conducted using SparQ enzymatic DNA fragmentation and illumina compatible iTru adapters and were sequenced using an illumina paired-end P3 300 cycle flow cell platform (Illumina, Inc., San Diego, CA, USA).

Paired-end reads were mapped to either our *S. exigua* unphased hybrid XY genome or the *S. purpurea* version 5 genome assembly[35] with the W chromosome (without Chr15Z) using bwa mem v0.7.17-r1188[66]. Mapping output was converted into BAM files following the Broad Gold Standard methods for variant processing[67,68]. The HaploCaller module from GATK v.4.0.12.0[69] was employed to call variants under parameters of allowing minimum heterozygosity 0.015 for SNP and 0.01 for indels. The raw VCF file was generated by the GenotypeGVCFs module. To remove low quality data, filtering was performed under

criteria of MQ < 15, SOR > 4, QD < 20, FS > 60, MQRankSum < −10, ReadPosRankSum < -2, ReadPosRankSum > 2, and DP between 6 and 100. SNP loci with at least 75% genotypes in filtered VCF files of each species were used for downstream analyses. Sex was included and tested as a binary phenotype using the GWAS pipeline using PLINK v1.90b4[70].

### Phasing sex-linked haplotypes in S. exigua

To phase sex-linked haplotypes, we grew 48 half-sibs from seeds collected from the same *S. exigua* female that was used for genome assembly (#SE916F). The same extraction, sequencing, mapping, and SNP calling pipelines as detailed above were used. Sexes of each half-sib were determined using sex-associated SNPs identified in our mapping study (see above). Y chromosome-specific alleles in the *S. exigua* SDR were identified as alleles in the male progeny that were not present in the mother or the female progeny. X-alleles were identified as the alternate alleles in the female half-sib progeny at the same loci where the Y-alleles were identified. X- and Y-alleles were confirmed as sex-linked when they were sex-linked in our GWAS analysis.

Flanking regions (150 bp) of Y-linked SNPs that mapped onto the *S. purpurea* 15 W chromosome were extracted to align in *S. purpurea* 15Z and *S. exigua* Chr15 to decide the movement of sex-linked region among different sex chromosomes using BLASTN[71]. Plotting the positions and connection between pairs of flanking regions were performed using ggplot2 package v3.4.0 in R v.4.2.1[72].

### Identification of Y-specific contigs

Y-specific contigs from our *S. exigua* genome assembly were identified by comparing sequence coverage on Chr15 and all unassembled scaffolds using our illumina sequencing reads from sequence capture for 24 males and 24 females. All the reads were filtered for a minimum Mapping Quality of 30. Y-specific regions are expected to be absent from females. We identified one candidate Y-specific region and used it to identify and extract a Y-specific contig from the male Nanopore assembly. Additional alignments were performed against the female genome assembly to confirm that the Y-specific regions were absent from females. Synteny maps between candidate and sex chromosomes were generated using R package ggplot2[72].

### Identifying RR17 partial duplicates

*RR17* gene (P. *trichocarpa*: XM_002325453.3) copies were queried at both the chromosome and contig level in our *S. exigua* assembly, in *S. purpurea* Chr15 Z and Chr15W, and on Chr19 of both *S. exigua* and *S. purpurea* using BLASTN v. 2.6.0 with -wordsize 8[71]. Different partial duplicate copies on Chr15Y of *S. exigua* and Chr15Z of *S. purpurea*, and the full-length *RR17* genes from Chr 15W of *S. purpurea* and Chr19 of *P. trichocarpa* (an outgroup) were aligned by ClustalW, respectively[73]. Phylogenies of each of the partial duplicates were reconstructed by MEGA11 using Maximum Likelihood with 100 bootstraps[74].

### Reporting summary

Further information on research design is available in the Nature Portfolio Reporting Summary linked to this article.

## Data availability

The Oxford Nanopore (ONT) reads generated in this study are deposited in the NCBI BioProject under accession number PRJNA1010806. The Illumina and Hi-C interaction mapping reads generated in this study for genome assembly polishing are deposited in the NCBI BioProject under accession number PRJNA1010212. The population sequence capture array Illumina reads generated in this study have been deposited in the NCBI BioProject under accession number PRJNA1009225. The genome assembly of male and female *Salix exigua* have been deposited in the NCBI BioProject under accession number PRJNA1009227 and PRJNA1009230.

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

## Acknowledgements
We thank Ashmita Khanal, Ken Keefover-Ring, and Matt Johnson for helpful discussion and comments. This research was supported by grants from the US National Science Foundation (1542509 to S.D., 1542599 to M.S.O., 1542486 to L.B.S.) and the National Natural Science Foundation of China (31561123001 to J.L.).

## Author contributions
M.S.O., J.L., L.B.S., S.P.D., and T.M. conceived of the project and developed resources; N.H., B.J.S., G.F., and M.S.O. designed and developed experiments and the mapping pipeline; N.H., B.J.S., M.G., G.F., D.G., H.H., and M.S.O. identified and sampled plant materials; D.W., J.L., and T.M. developed the RR17 analysis pipeline; B.H., L.B.S., and S.P.D. contributed and analyzed Salix purpurea diversity data; N.H. and M.S.O. drafted of the manuscript with significant contributions from B.J.S., M.G., D.G., J.L., L.B.S., S.P.D., and T.M.

## Competing interests
The authors declare no competing interests.
