## [Peer Review File · Nature Communications]

REVIEWER COMMENTS

Reviewer #1 (Remarks to the Author):

Hu et al., contribute their study of "An unusual origin of a ZW sex chromosome system". Transition in sex determination is an important and interesting topic, especially the genetics/genomics of a fast transition. Amazing advancement in genome sequencing provides unprecedented opportunities in this area. In this study, a new model of fast sex determination in *Salix* species, specifically origin model of a ZW sex chromosome system, is raised by exploration of whole genome sequence data.

This paper reveals evidences from genomic data analysis, raised novel model of sex determination origin. But limitation is seen in primary data preparation and paper structure. I suggest to accept after a major revision.

major points:

1. One my big concern is the quality of genome assembly (in fact two, one male and one female) this study provided and explored. It is really hard to imagine you just got such a low continuity (reflected in N50) with such high sequencing depth (>100x) of ONT. Even though it may not influence the key analyses and further the results and conclusions, I think it is not a good practice to publish them without a thoroughgoing check and improvement.
2. Core hypothesis is raised partially by tracing the homology of SDR of Z and W in *S. purpurea* to Y in *S. exigua*. I am thinking why you did not do a full phylogenetic tree (like you provided in F.g 4c) by including RR17 and partial duplicates from more species especially those of *Salix*? And also tree of only the partial duplicates from multiple species.
3. Also, one my concern and also as a suggestion. Why you did not sequence alignment of potential SDR and flanking sequences between Z and W in *S. purpurea* to Y and also X in *S. exigua*? This is still valuable to clarify thing important to your key finding.
4. In results part, you may need an enrichment, by adding sample preparation, genome sequencing/assembly, quality control of the assembly not only with BUSCO, SNP calling and manipulation, quality control of capture sequencing and variant calling. Even though these look tiny to the major point you raised, they are truly crucial.

minor points:

1. Figure S1 is not referred in the main text
2. Figure S2, What exactly the information you want to bring? Could you highlight the specific information by zoom-ins, underscores or squares? And further provide more description in the main text.
3. Table S2, I found "RC_contig_511_:::fragment_2" but no "Contig 511". They are different by name/ID.
4. Table S3, how you define "Direction"? why it is in "+" and "-"?
5. line 118, "Hi-C assembly"? It is better to be "Hi-C assisted assembly" or "Hi-C determined assembly" or any others, but not this "Hi-C assembly". How do you think of this?
6. Figure 4, caption, please be specific. In "Red shows copies in *S. exigua*. Purple shows copies in *S. purpurea*", replace "copies" with "partial duplicate copies".
7. Figure 5, please further improve the caption. What is the blue boxes? Yes, they look like the homologous sequences, but be specific.

8. Table S4, provide full description to each columns, even though BLASTn output is clear to many readers. Do you mean the RR17 sequence you used in this BLAST is from *Populus trichocarpa*? Be specific again to many such important details.

9. Text is not read smoothly, please improve your expression.

Reviewer #2 (Remarks to the Author):

In the manuscript "An unusual origin of a ZW sex chromosome system" by Hu et al., the authors use a new chromosome-level genome assembly together with sequence capture genotyping of *Salix exigua* to identify and characterize its sex-determining region. They demonstrate that *S. exigua* exhibits an XY system of sex determination, which may function via Y-specific partial duplicates of a response regulator gene, similar to some other willow and poplar species. Using sequence alignments between *S. exigua* Y- and *S. purpurea* Z- and W- chromosomal sequences and segregation pattern in an *S. exigua* half-sib progeny, the authors conclude that both the Z and the W originated from the *S. exigua* Y chromosome while the X chromosome was lost.

While I found the GWAS results and phylogenetic analyses of the RR gene sequences convincing, I was missing some more thorough analyses of the two haplotypes. Since the genome of a male individual was assembled, it should be possible to directly compare the X and the Y haplotypes. This would allow analyzing in more detail possible X-specific genes or sequences, which I consider important for the claim that the X was lost during the transition to female heterogamety (see below).

With the current analyses, I find this claim questionable. While it appears clear that part of the male-specific sequence has been translocated to what evolved into the Z-chromosomal part of the *S. purpurea* SDR, the relationships between X, Y and W are less obvious to me. The main reason for this is the segregation analyses. Using an *S. exigua* half-sib progeny, the authors define Y-alleles and X-alleles. However, I failed to understand the according methods, and can therefore not judge whether the conclusions are sound.

Specifically, I did not understand how the phasing of the sequence capture SNPs from the half-sib family was used to define Y-alleles (of which 198 are discussed in the results section). In my understanding, a Y-allele should be specific to males, that is males should be heterozygous at that position while females should be homozygous. However, the authors write that "When the mother was 0/0, Y-alleles could be identified when male progeny were 0/0, 0/1, or 0/2" (L344). When the mother is 0/0 and the male progeny is 0/0, which allele is supposed to be the Y-allele? Also, when the mother is 0/1 and the male progeny is 0/0 or 1/1, how can you possibly define a Y-specific allele (and the according X-allele)?

From the 198 "Y-alleles" that map to the *S. purpurea* SDR, only 30 that are also associated with sex when mapping the *S. exigua* data to the *S. purpurea* reference were analyzed in more detail. Which alleles of the remaining 168 "Y-alleles" are present on the Z or W? And what about the sex-linked alleles of *S. exigua* that do not map to the *S. purpurea* SDR?

While I understand that the story is about transitions between heterogametic systems, it would be interesting to learn more about the putative function of the characterized sequences. For example, readers may wonder how the partial and complete RR gene sequences are expressed in female and male plants? This might make the subsequent analyses even more interesting.

Finally, when considering that only part of chromosome 15 is responsible for sex determination, I find it difficult to talk about "loss of the X chromosome". Therefore, I would suggest to make clear that only part of the chromosome is involved in sex determination and transitions do not involve the entire chromosome. This may be most relevant for Fig. 6. Also, it may be relevant for the conclusions regarding the transitions. If in fact the Y-chromosomal part of *S. exigua* has been translocated to the middle of chromosome 15 to become the Z- and W-chromosomal part in the ZW system, the X-chromosomal sequences of *S. exigua* might still be found at the genomic

position of the *S. exigua* SDR in *S. purpurea*.

Specific comments:

L18: why do you start the abstract with "transitions in the heterogamety are rare", if in the introduction you cite dozens of papers demonstrating such transitions in many different species?

L20 and 48-50: I am not convinced that X>W and Y>Z transitions would make more sense and are thus more likely to occur. In an active Y system, the X does not need to carry any feminizing genes. A dominant masculinizing gene (such as SRY) on the Y is sufficient. Also, the Z does not need to carry any masculinizing genes.

L113-115: from Fig. 5 I cannot see less synteny with *S. purpurea* Chr15W compared to 15Z. Could you perhaps specify the difference numerically? Perhaps also the missing synteny with female *S. exigua*?

L116-118: are those two copies orthologs of the two full-length copies on Chr19 of *S. purpurea*, or are they more closely related to the full-length copies on *S. purpurea* Chr15W? Could you add the *S. purpurea* Chr19 RR sequences to the phylogeny in Fig. 4c?

L132-137: I do not understand the difference between mapping "to the *S. purpurea* genome that included only the 15W chromosome" (L134) and mapping "to the *S. purpurea* reference genome with the 15W chromosome" (L137).

L174-175: I failed to reconcile this statement with the data presented in Fig. 8 of ref. 34 from the same authors (Zhou et al. 2020). Expression of Sapur.15 W073500 appears to be strongly female-biased in panels a and b. How do these data fit the new claim that "RR17 does not exhibit sex-biased expression in *S. purpurea*"?

L186: I do not believe that the W will evolve dominance over the Y in this specific system. If the masculinizing factors were not lost, the W may not be dominant. Additionally, see comment on L20

Minor comments:

L57-58: it would be easier for the reader if you named the non-dioecious species

L60: do you really want to cite Yin et al. (2008), which suggested a ZW system for *P. trichocarpa*? Instead, you could consider adding a citation reporting the XY system of *P. tremula*

L61: what about *P. adenopoda* (Kim et al. (2021), <https://doi.org/10.2478/sg-2021-0012>)?

L163-166: why is this considered interesting? I would consider it a logical consequence of the switch from the XY to the ZW system. The sRNAs appear to act as dominant masculinizing factors in other species. Such a factor would not be compatible with a ZW system.

Reviewer #3 (Remarks to the Author):

This is a fairly straightforward study examining the origins of sex determination turnover among *Salix* species. The phylogeny combined with documentation of the chromosomal origins of the SD regions are interesting. While the study does not make major strides towards our understanding of molecular or evolutionary mechanisms, it does document an interesting case of heterogametic turnover.

However, I am not as convinced by the case for 'Y origins' for Z and W. The argument currently is resting on a very small number of SNP patterns, without an explicit gene tree reconstruction with all of the species included. Why not reconstruct an explicit phylogeny, including X, Y, Z, W and outgroup sequences, to more directly document the Y origin of Z and W, rather than simply relying on SNP segregation patterns? This would be important to back up the claims of the study.

RESPONSE TO REVIEWERS' COMMENTS

Reviewer #1 (Remarks to the Author):

Hu et al., contribute their study of "An unusual origin of a ZW sex chromosome system". Transition in sex determination is an important and interesting topic, especially the genetics/genomics of a fast transition. Amazing advancement in genome sequencing provides unprecedented opportunities in this area. In this study, a new model of fast sex determination in *Salix* species, specifically origin model of a ZW sex chromosome system, is raised by exploration of whole genome sequence data.

This paper reveals evidences from genomic data analysis, raised novel model of sex determination origin. But limitation is seen in primary data preparation and paper structure. I suggest to accept after a major revision.

Thank you for your comments on our manuscripts. We have incorporated all of your concerns in our manuscript.

major points:

1. One my big concern is the quality of genome assembly (in fact two, one male and one female) this study provided and explored. It is really hard to imagine you just got such a low continuity (reflected in N50) with such high sequencing depth (>100x) of ONT. Even though it may not influence the key analyses and further the results and conclusions, I think it is not a good practice to publish them without a thoroughgoing check and improvement.

Our assembly scaffolding process relied on Hi-C interactions more than on the ONT reads. The "low N50" is purposely managed by FLYE assembler parameter. FLYE provides a choice between two modes of scaffolding: 1) a mode to self-scaffold based on ONT reads, and 2) a mode that scaffolds using Hi-C reads (when available) without first scaffolding using the ONT reads. Normally, using mode 1 will drastically increase the N50 beyond 1M. We completed an assembly using ONT scaffolding and our N50 was ~1.3M, but we did not report this because we chose mode 2, which is recommended by the program authors when Hi-C interaction sequencing reads are available to assist the scaffolding process. We used mode 2 to avoid introducing errors from ONT scaffolding. In our first submission we reported only the N50 prior to scaffolding and with no self-scaffolding. We understand how this is confusing. After Hi-C scaffolding, our N50 reaches to ~10.3M, which is a chromosomal level assembly, and we now report this value in our manuscript.

We agree that our assembly of the female individual needed improvement. We further assembled the female genome with ONT scaffolding and now report the results of this assembly, which had an N50 of 655k and was unfortunately not as good as the male. The alignment of these reads to the male assembly shows that we have assembled the majority of chr7 (X chromosome) (See updated Figure 2).

2. Core hypothesis is raised partially by tracing the homology of SDR of Z and W in *S. purpurea* to Y in *S. exigua*. I am thinking why you did not do a full phylogenetic tree (like you provided in F.g 4c) by including

RR17 and partial duplicates from more species especially those of *Salix*? And also tree of solely the partial duplicates from multiple species.

We added Figure 4b, which is a phylogeny based on the 18 homologous genes on Z, W, X, and Y that contain at least one SNP in our population data. This phylogeny shows that the Z & W chromosomes were derived from the Y.

We also updated former figure 4c (now figure 6). The phylogeny of all RR17 partial duplicates across species is hard to interpret since many of these partial duplicates are duplications from different regions of RR17 and do not share homology. In response to your concerns, we created phylogenies in Figure 6 that include all homologous regions of the partial duplicates and full length copies of RR17 from *S. purpurea*, *S. exigua* and the outgroups. These two *Salix* species represent the two major clades included in the 15XY to 15ZW shift. Another study of a well-assembled species with Chr15ZW as its sex determination system, *S. viminalis*, failed to find RR17 partial duplicates (unpublished data), so we did not add them here. We feel that this level of taxonomic sampling provides a robust answer to our main question of whether the RR17 duplicates are homologous and arose once prior to the divergence between *S. exigua* and *S. purpurea*, or if they arose twice independently. Our results in the paper support the former hypothesis (arose once).

3. Also, one of my concerns and also as a suggestion. Why you did not sequence alignment of potential SDR and flanking sequences between Z and W in *S. purpurea* to Y and also X in *S. exigua*? This is still valuable to clarify things important to your key finding.

Thank you for the suggestion. We added a synteny analysis across Z, W, X, and Y chromosome based on our new assembly of X contigs in Figure 2 (X, Y, and Z) and Figure S3 (Z and W).

4. In results part, you may need an enrichment, by adding sample preparation, genome sequencing/assembly, quality control of the assembly not only with BUSCO, SNP calling and manipulation, quality control of capture sequencing and variant calling. Even though these look tiny to the major point you raised, they are truly crucial.

We added more information about our sequencing and variant calling results which we hope will address your concerns.

minor points:

1. Figure S1 is not referred in the main text

Done.

2. Figure S2, What exactly the information you want to bring? Could you highlight the specific information by zoom-ins, underscores or squares? And further provide more description in the main text.

We updated the figure legend to clarify interpretation of the figure. The main point of this Hi-C interaction map is to support our confidence in the assembly. The figure shows the level of the chromatin interaction signals between two segments. More red color represents higher interactions, which indicates a physically closed DNA segment. A Hi-C interaction block normally represents a physical linkage group that have low chromatin interactions with other groups.

3. Table S2, I found "RC_contig_511_:::fragment_2" but no "Contig 511". They are different by name/ID.

When 3D-DNA assembler identifies a Nanopore contig as misjoined and corrects it so that only a portion of the Nanopore contig is included in the final assembly, the assembly flags it as "...fragment_#". Although most of Contig 511 is syntenic with Chr15 in *S. purpurea*, the first 200kbp of Contig 511 is not syntenic with Chr15 in *S. purpurea*. Also, only the second half of Contig 511 is male specific. We added a description of the meaning of "...fragment_#" to the table legend.

4. Table S3, how you define "Direction"? why it is in "+" and "-"?

It is the direction how the DNA segment aligned to each other and is relative to the direction of the *S. purpurea* chromosomes. We added a short statement in the table legend to clarify these symbols.

5. line 118, "Hi-C assembly"? It is better to be "Hi-C assisted assembly" or "Hi-C determined assembly" or any others, but not this "Hi-C assembly". How do you think of this?

We agree with this and changed it to Hi-C assisted assembly.

6. Figure 4, caption, please be specific. In "Red shows copies in *S. exigua*. Purple shows copies in *S. purpurea*", replace "copies" with "partial duplicate copies".

Because we rearranged some figures, this is now figure 5. We provided more detail in the legend as requested.

7. Figure 5, please further improve the caption. What is the blue boxes? Yes, they looks like the homologous sequences, but be specific.

This is now Figure 4a. The blue boxes represent the homologous flanking regions around selected SNP. All of them are 151bp long. We updated it in the figure legends.

8. Table S4, provide full description to each columns, even though BLASTn output is clear to many readers. Do you mean the RR17 sequence you used in this BLAST is from *Populus trichocarpa*? Be specific again to many such important details.

We added a description of each column above the table. And we updated the title to specify this is BLASTn results using RR17 intact gene from *P. trichocarpa*.

9. Text is not read smoothly, please improve your expression.

We revised the text throughout to improve the readability.

Reviewer #2 (Remarks to the Author):

In the manuscript “An unusual origin of a ZW sex chromosome system” by Hu et al., the authors use a new chromosome-level genome assembly together with sequence capture genotyping of *Salix exigua* to identify and characterize its sex-determining region. They demonstrate that *S. exigua* exhibits an XY system of sex determination, which may function via Y-specific partial duplicates of a response regulator gene, similar to some other willow and poplar species. Using sequence alignments between *S. exigua* Y- and *S. purpurea* Z- and W- chromosomal sequences and segregation pattern in an *S. exigua* half-sib progeny, the authors conclude that both the Z and the W originated from the *S. exigua* Y chromosome while the X chromosome was lost.

While I found the GWAS results and phylogenetic analyses of the RR gene sequences convincing, I was missing some more thorough analyses of the two haplotypes. Since the genome of a male individual was assembled, it should be possible to directly compare the X and the Y haplotypes. This would allow analyzing in more detail possible X-specific genes or sequences, which I consider important for the claim that the X was lost during the transition to female heterogamety (see below).

Thank you for your comments on our manuscripts. To answer your concern about comparing X and Y haplotypes, we extracted genes from the X and Y haplotypes that contained at least one SNP that can be phased to reconstruct the phylogeny. The phylogeny of these genes is shown in fig 4b and has the same result and interpretation as the phylogeny when we used only the SNPs. We updated our figures and text to incorporate this new result.

With the current analyses, I find this claim questionable. While it appears clear that part of the male-specific sequence has been translocated to what evolved into the Z-chromosomal part of the *S. purpurea* SDR, the relationships between X, Y and W are less obvious to me. The main reason for this is the segregation analyses. Using an *S. exigua* half-sib progeny, the authors define Y-alleles and X-alleles. However, I failed to understand the according methods, and can therefore not judge whether the conclusions are sound.

Specifically, I did not understand how the phasing of the sequence capture SNPs from the half-sib family was used to define Y-alleles (of which 198 are discussed in the results section). In my understanding, a Y-allele should be specific to males, that is males should be heterozygous at that position while females should be homozygous. However, the authors write that “When the mother was 0/0, Y-alleles could be identified when male progeny were 0/0, 0/1, or 0/2” (L344). When the mother is 0/0 and the male progeny is 0/0, which allele is supposed to be the Y-allele? Also, when the mother is 0/1 and the male progeny is 0/0 or 1/1, how can you possibly define a Y-specific allele (and the according X-allele)?

We now see that our presentation of these methods was confusing and have revised this presentation. The 30 alleles that we identified as Y-specific were identified as present in the male progeny, but not present in the mother or any female progeny.

From the 198 “Y-alleles” that map to the *S. purpurea* SDR, only 30 that are also associated with sex when mapping the *S. exigua* data to the *S. purpurea* reference were analyzed in more detail. Which alleles of the remaining 168 “Y-alleles” are present on the Z or W? And what about the sex-linked alleles of *S. exigua* that do not map to the *S. purpurea* SDR?

In our previous analysis, we identified alleles that were on the Y chromosome, but not necessarily Y-specific. We recognize that our description was confusing and unnecessary given the goals of our questions. Our manuscript now only focuses on the 30 loci for which we have accurate phasing of the X- and Y-alleles.

While I understand that the story is about transitions between heterogametic systems, it would be interesting to learn more about the putative function of the characterized sequences. For example, readers may wonder how the partial and complete RR gene sequences are expressed in female and male plants? This might make the subsequent analyses even more interesting.

We agree that the function of RR17 partial duplicates would be interesting to study. However, expression analyses are beyond the scope (and budget) of our current study. A recent published paper in *S. arbutifolia* found a signal of siRNA expression only in males, which is indirect evidence.

Finally, when considering that only part of chromosome 15 is responsible for sex determination, I find it difficult to talk about “loss of the X chromosome”. Therefore, I would suggest to make clear that only part of the chromosome is involved in sex determination and transitions do not involve the entire chromosome. This may be most relevant for Fig. 6. Also, it may be relevant for the conclusions regarding the transitions. If in fact the Y-chromosomal part of *S. exigua* has been translocated to the middle of chromosome 15 to become the Z- and W-chromosomal part in the ZW system, the X-chromosomal sequences of *S. exigua* might still be found at the genomic position of the *S. exigua* SDR in *S. purpurea*.

We agree to change the expression of our conclusion to “loss of the X-linked allele”. And we specified the changes are only limited to SDR regions between two species. Translocation is one hypotheses for the movement of these RR17 partial duplicates. In Zhou al., 2020, they also found a large-scale palindrome structure on Chr15 of *S. purpurea*, indicating a chromosomal rearrangement occurred in its ancestors.

The RR17 partial duplicates clustered in a very small region (10kbp), so it is possible no functional genes were translocated with the RR17 partial duplicates.

Specific comments:

L18: why do you start the abstract with “transitions in the heterogamety are rare”, if in the introduction you cite dozens of papers demonstrating such transitions in many different species?

Good point! We no longer start the abstract with this statement. Nonetheless, most of sex chromosome transitions do not shift heterogamety.

L20 and 48-50: I am not convinced that X>W and Y>Z transitions would make more sense and are thus more likely to occur. In an active Y system, the X does not need to carry any feminizing genes. A dominant masculinizing gene (such as SRY) on the Y is sufficient. Also, the Z does not need to carry any masculinizing genes.

We have toned down our statement. However, we still maintain that since the X & W are associated with feminization and the Y & Z are associated with masculinization, it seems naively reasonable that X>W and Y>Z transitions would be likely. We are not claiming that this is the way it occurs, only pointing out that this is a likely outcome given the sex associations with the various chromosomes.

L113-115: from Fig. 5 I cannot see less synteny with *S. purpurea* Chr15W compared to 15Z. Could you perhaps specify the difference numerically? Perhaps also the missing synteny with female *S. exigua*?

The previous Figure 5 is now Figure 4a because we rearranged the presentation of the results. To answer your question, we added a protein coding gene synteny between Contig 511 to both Chr15Z and Chr15W of *S. purpurea* in Table S3. The synteny showed that there are 16 genes shared between Contig 511 to Chr15Z while only 10 genes shared between Contig 511 and Chr15W, indicating a lower similarity compared to Chr15Z. Also, for your second question, the previous Figure 3 is now part of Figure 2. Now, Figure 2a showed the synteny between contig 511 and female Chr15 (X-contigs).

L116-118: are those two copies orthologs of the two full-length copies on Chr19 of *S. purpurea*, or are they more closely related to the full-length copies on *S. purpurea* Chr15W? Could you add the *S. purpurea* Chr19 RR sequences to the phylogeny in Fig. 4c?

We added the full length copies of RR17 gene from both *S. exigua* and *S. purpurea* genome to our phylogeny.

L132-137: I do not understand the difference between mapping “to the *S. purpurea* genome that included only the 15W chromosome” (L134) and mapping “to the *S. purpurea* reference genome with the 15W chromosome” (L137).

Thank you for pointing it out. It should refer to the same thing. We edited it to make it consistent.

L174-175: I failed to reconcile this statement with the data presented in Fig. 8 of ref. 34 from the same authors (Zhou et al. 2020). Expression of Sapur.15 W073500 appears to be strongly female-biased in panels a and b. How do these data fit the new claim that “RR17 does not exhibit sex-biased expression in *S. purpurea*”?

Sorry for mistakenly stating the wrong information. We fixed our statement.

L186: I do not believe that the W will evolve dominance over the Y in this specific system. If the masculinizing factors were not lost, the W may not be dominant. Additionally, see comment on L20

Some mechanisms of dominance have been proposed in models including that the dominance on the W could be a dosage effect that overcame the old masculinizing factors on ancient Y, or it could result from a sex-specific allocation factor that reallocates resources to females from males. However, none of these mechanisms have been proven in this system. We also added supporting evidence on L195.

Minor comments:

L57-58: it would be easier for the reader if you named the non-dioecious species

Done.

L60: do you really want to cite Yin et al. (2008), which suggested a ZW system for *P. trichocarpa*? Instead, you could consider adding a citation reporting the XY system of *P. tremula*

Done. We removed the misleading citation and added XY report of *P. tremula*.

L61: what about *P. adenopoda* (Kim et al. (2021), <https://doi.org/10.2478/sg-2021-0012>)?

RR17 gene function is clearer in *Populus* species than *Salix*. We will cite this study.

L163-166: why is this considered interesting? I would consider it a logical consequence of the switch from the XY to the ZW system. The sRNAs appear to act as dominant masculinizing factors in other species. Such a factor would not be compatible with a ZW system.

This sentence is simply stating how the W chromosome arose.

Reviewer #3 (Remarks to the Author):

This is a fairly straightforward study examining the origins of sex determination turnover among *Salix* species. The phylogeny combined with documentation of the chromosomal origins of the SD regions are interesting. While the study does not make major strides towards our understanding of molecular or evolutionary mechanisms, it does document an interesting case of heterogametic turnover.

However, I am not as convinced by the case for 'Y origins' for Z and W. The argument currently is resting on a very small number of SNP patterns, without an explicit gene tree reconstruction with all of the species included. Why not reconstruct an explicit phylogeny, including X, Y, Z, W and outgroup sequences, to more directly document the Y origin of Z and W, rather than simply relying on SNP segregation patterns? This would be important to back up the claims of the study.

The results rely on 18 genes with X and Y-linked alleles. The requested phylogeny is presented in Figure 4b.

REVIEWERS' COMMENTS

Reviewer #1 (Remarks to the Author):

In this revision, I am generally satisfied with the changes. Though the genome assembly is not so clear for me and the genomic analyses still could be improved further, I do not have follow-up comments at this moment. I will recommend to accept it for publication, if it meets the journal's requirements.

Reviewer #2 (Remarks to the Author):

The authors have addressed all of my previous comments. I enjoyed reading the revised version of the manuscript.

The only point that I could still not follow is L125-126 "Contig 511 exhibited partial synteny with a large section of the sex determination region of *S. purpurea* Chr15Z and minimal synteny with *S. purpurea* Chr15W (Fig. 2a, Table S3)." In the part of Table S3, where Chr15Z and Chr15W are compared (annotated genes shared with contig 511, if I understood that part correctly), the total gene length is 23k for Chr15Z and 37k for Chr15W. So where is the minimal synteny with *S. purpurea* Chr15W compared to Chr15Z?

Just a small comment on L161-165. This makes a lot of sense if the partial duplicates are/were functional. To silence the RR17 gene, I would expect that the transcriptional start site of the gene should be targeted for methylation. The regions further downstream in contrast may never have played any functional role.

RESPONSE TO REVIEWERS' COMMENTS

Reviewer #1 (Remarks to the Author):

In this revision, I am generally satisfied with the changes. Though the genome assembly is not so clear for me and the genomic analyses still could be improved further, I do not have follow-up comments at this moment. I will recommend to accept it for publication, if it meets the journal's requirements.

Thank you for your careful review.

Reviewer #2 (Remarks to the Author):

The authors have addressed all of my previous comments. I enjoyed reading the revised version of the manuscript.

The only point that I could still not follow is L125-126 “Contig 511 exhibited partial synteny with a large section of the sex determination region of *S. purpurea* Chr15Z and minimal synteny with *S. purpurea* Chr15W (Fig. 2a, Table S3).” In the part of Table S3, where Chr15Z and Chr15W are compared (annotated genes shared with contig 511, if I understood that part correctly), the total gene length is 23k for Chr15Z and 37k for Chr15W. So where is the minimal synteny with *S. purpurea* Chr15W compared to Chr15Z?

Thank you. We added additional information about the synteny between Contig511 to Chr15W that included the cumulative collinear length in our Supplementary Data 3 (formerly Table S3). The cumulative syntenic length when aligned to Chr15W was less than when aligned to Chr15Z, although “minimal” is not the best way to describe synteny with Chr15W. We also changed the text to read: “Contig 511 exhibited partial synteny with a large section of the sex determination region of *S. purpurea* Chr15Z and less synteny with *S. purpurea* Chr15W (Fig. 2a, Supplementary Data 3).”

Just a small comment on L161-165. This makes a lot of sense if the partial duplicates are/were functional. To silence the RR17 gene, I would expect that the transcriptional start site of the gene should be targeted for methylation. The regions further downstream in contrast may never have played any functional role.

Thank you. This is a good observation that we also noticed. Since we have no direct data regarding the functions of the RR17, we decided to not mention this speculative hypothesis and leave it for future research to address.